# Revealing the Effectiveness of Fisheries Policy: A Biological Observation of Species *Johnius belengerii* in Xiamen Bay

**Liang-Min Huang** [1,2], **Jia-Qiao Wang** [1,2], **Yi-Jia Shih** [1,2], **Jun Li** [1,2] and **Ta-Jen Chu** [1,2,*]

[1] Fisheries College, Jimei University, Xiamen 361021, China; lmhuang@jmu.edu.cn (L.-M.H.);
skyofstar1@jmu.edu.cn (J.-Q.W.); eja0313@gmail.com (Y.-J.S.); lijun1982@jmu.edu.cn (J.L.)
[2] Fujian Provincial Key Laboratory of Marine Fishery Resources and Eco-Environment, Jimei University,
Xiamen 361021, China
[*] Correspondence: chutajen@jmu.edu.cn

**Abstract:** The rapid development of China's economy has brought tremendous pressure to the marine ecosystem, and about 57% of marine fish populations have been overexploited or collapsed. A series of fisheries policies have been implemented successively to improve the decline of resources. Over the past decade, the fisheries sector has particularly increased focus on resource and ecosystem sustainability, which has led to the wider use of stock management policies in China. Therefore, fishery resource assessment is crucial, such as assessing the long-term changes in biological information. This study is based on biological characteristics of *Johnius belengerii* captured by bottom trawls in Xiamen Bay during two periods, beginning in 2006 and 2016. Length composition, length–weight relationship, growth, mortality, sexual maturation, and feeding intensity were analyzed. The changes in biological characteristics show that there is a phenomenon of improvement in the later period, of which the changes are closely related to a series of fisheries management strategies, such as setting closed fishing periods and non-fishing areas, and establishing fishery restoration marine protected areas. This result seems to reveal the effectiveness of a long-term series of fisheries policies. It can provide an important basis and visibility for management effectiveness.

**Keywords:** *Johnius belengerii*; Xiamen Bay; biological characteristics; fisheries policy

## 1. Introduction

In recent years, the total global fisheries catch has reached its highest level in history at 96.4 million tons, an increase of 5.4% over the past average. In terms of the fishing fleet, fishery employees, and marine fishing output, China has the world's largest fishery industry [1]. In 2018, China accounted for 18.9% of global fishing vessels and 15% of marine catches (FAO, 2020) [2]. However, in many of the world's offshore and pelagic fisheries, nearly 90% of marine fish stocks have been fully exploited, overexploited, or depleted [2–5]. At the same time, the rapid development of the Chinese economy has put tremendous pressure on the marine ecosystem, and approximately 57% of marine fish populations have been overexploited or collapsed [6]. Protecting offshore fishery resources, reducing fishing intensity, and enhancing fishing capacity are the core requirements for the sustainable development of marine fisheries [7,8]. Therefore, fisheries policies must be developed to implement the core requirements of sustainable development. Furthermore, the analysis and evaluation of fisheries policies is an important subject and task in the supervision of offshore capture fisheries.

Since the 1990s, with the continuous reform of China's economic system and the implementation of the policy of opening up to the outside world, China's fishery industry has developed rapidly. The total output of aquatic products increased from $1237 \times 10^4$ tons in 1990 to $3601 \times 10^4$ tons in 1997 [9]. Liu [9] reported that offshore marine fishing output has been maintained at a high level, resulting in a decline in fishery resources [10]. Some studies suggest that overfishing, marine pollution, beach reclamation destruction, and

habitat destruction by coastal engineering are the four main reasons for the decline in fishery resources [11]. Wang and Liu (2009) mentioned that, especially after the signing of the China-Japan Fisheries Agreement, the China-Korea Fisheries Agreement, and the China-Vietnam Northern Gulf Fisheries Agreement, coastal fishing grounds have shrunk significantly in China [12]. To protect offshore fishery resources, the government has put forward corresponding policies, including the implementation of policies for reducing boats and changing the industry, formulating an evaluation plan for fishing intensity control, adjusting the industrial structure of fishing areas, and strictly distinguishing the types of boat reductions [10]. Zheng [13] mentioned that fisheries management policies before 1999 did not have a significant effect on the effective control of fishing capacity. However, from 1999 to 2004, because the government attached great importance to the reduction in fishing capacity, a series of fisheries policies were implemented, resulting in relatively evident comprehensive results [13]. Zheng et al. [14] mentioned that strengthening the control over fishing capacity has become an urgent task for fishery managers. It is beneficial to restore and conserve offshore fishery resources and ensure the sustainable development of marine capture fisheries [14]. Some management systems for the protection of fishery resources implemented in terms of controlling fishing capacities, such as the fishing license system, fishing moratoriums in the off-season, fishermen changing jobs, controls of fishery inputs and outputs, and fishing quotas, have become normal affairs [15].

Collecting biological and fishery data is crucial for stock assessments. These data provide useful information for fishery managers, including stock age structure, age at first spawning, fecundity, male-to-female ratio, natural mortality (M), fishing mortality (F), growth rate, and spawning behavior [16–18]. They include details on key habitats, migration habits, food preferences, and estimates of total biomass [19]. The evaluation of the effect of fisheries policy is an important link in the restoration of fisheries, and these procedures could provide an option for the management of the utilization and resource conservation of sustainable fisheries [20–22]. Some countries carry out fishery development projects every year to assess whether the policy is working. However, because their goals vary, they go beyond a limited focus on fisheries management and ecosystems. This makes it difficult to assess policy contributions, especially in most data-poor countries [22]. Therefore, there is a need for more connections and proof of biological information. However, very few studies have conducted relevant evaluations of the fisheries policies.

Sciaenids are important fishes with a global capture fishery production of approximately 1.7 million tons annually over the past decade (FAO, 2019) [23]. Some sciaenids form large aggregations for spawning, feeding, and over-wintering, which makes them vulnerable to overfishing [24]. One of the Sciaenids, *Johnius belengerii* (Cuvier, 1830), also called Belanger's croaker, Boulenger's croaker, or Belanger's jewfish, belongs to the superclass of Osteichthyes, family of Sciaenidae, and genus of Johnius [25]. They are distributed in the Indo-West Pacific, Pakistan, India, Sri Lanka, and via the East Indies to China [26], where they also inhabit coastal waters and estuaries [27]. In China, this species is commonly found near the shores of Zhejiang, Fujian, Guangdong, Hong Kong, and Taiwan [28]. It is of significant socio-economic value to humans and has historically been an important food source for many coastal communities. Many studies on *J. belengerii* focused on its various features, including its biological characteristics in Liusha Bay [29], the fisheries biology in the Ma'an Archipelago [26], feed patterns on invertebrates [30], its genetic diversity in the Beibu Gulf [31], and its temporal and spatial distribution in Xiamen Bay [32]. Zhong et al. (2000) reported that *J. belengerii* is the dominant species in Xiamen Bay [32]. Another species, *J. taiwanensis*, inhabits the same coastal waters [33].

Xiamen Bay is located on the southeast coast of Fujian Province and the sea area is approximately 390 square kilometers, with a total coastline of 234 km and 31 islands. Xiamen Bay is mainly an estuary of the Jiulong River, the second largest river in Fujian Province. It is rich in fishery resources and is a spawning ground for many fish and shrimps. There are various types of marine ecological environments, including nearly 2000 species of marine organisms and more than 60 species with high economic value [34]. In recent

decades, coastal areas have vigorously developed industries and constantly built ports, docks, roads, and bridges. The coastline was almost completely occupied. Human activities have brought some changes to the ecological environment, significantly reducing fishery resources [35].

Biological information and stock assessments are needed to manage the resources and sustainability of offshore fisheries in the Xiamen waters. The assessment of *J. belengerii* in Xiamen waters includes the investigation of parameters such as sex ratio, length–weight analysis, growth parameters, level of exploitation (natural mortality rate, fishing mortality rate, and total mortality rate), and feed intensity. In particular, two sets of biological information from 2006 to 2007 and from 2016 to 2017 are compared to evaluate the long-term effectiveness of a series of fisheries policies. Moreover, we aim to provide an idea regarding fisheries policy, reveal its effectiveness, and serve as a reference for formulating future resource management for offshore fisheries.

## 2. Biological Observation

### 2.1. Study Area

Xiamen Bay is an important economic zone in Xiamen and the whole of Fujian Province. The study area was located in Xiamen Bay ($117°50'–118°20$ E and $24°14'–24°42$ N), with a total area of 1281 km$^2$ (Figure 1). The entire area includes the eastern region (including Tong'an Bay, the Dadeng-Xiaodeng region, and Weitou Bay), which has a sandy substrate, and the western region (including Western Harbor and the Jiulong River Estuary), which is characterized by semi-closed estuaries [5,36]. Habitat types include mangrove wetlands and sandy beaches that have rich biodiversity [37]. This is because of the numerous ports, transport infrastructure, shipbuilding, and petrochemical industries. As a result, the region is also affected by intensive development activities, such as shipping, aquaculture, reclamation, and tourism. Unfortunately, these activities have resulted in several problems including marine biodiversity decline, habitat loss, and water pollution [38,39].

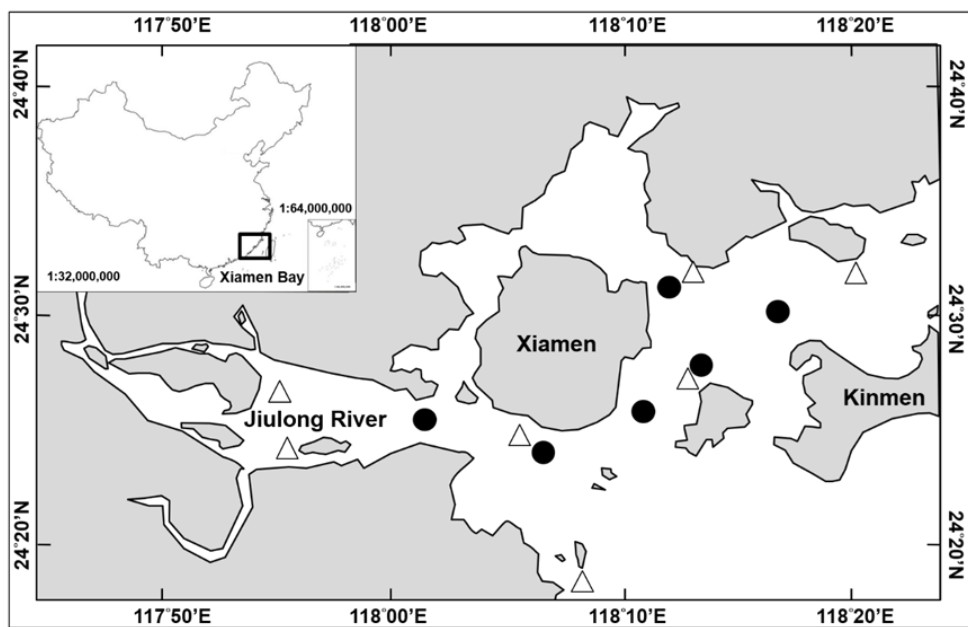

**Figure 1.** Map showing the location of the sample stations (Δ: 2006-2007; ●: 2016-2017) in Xiamen Bay.

Biological surveys were conducted at certain sites during these two periods (Figure 1). These sites are briefly described as follows. The estuary of the Jiulong River has a large freshwater exchange and mud sedimentation area, where the seabed substrata are muddy because of the material deposition by the river. Some sites are located within shallow waters, which face the open sea, where the mixing of substrate with mud–sand and continuous



water exchange occurs. Some sites are behind Kinmen and Xiaokinmen Islands, one of which is located at the mouth of Tongan Bay, which is a semi-enclosed water environment. A large amount of sludge is deposited in Tongan Bay behind the bay mouth because of seawater backflow.

### 2.2. Biological Sampling

Field surveys were conducted in 2006–2007 and 2016–2017 to explore the *J. belengerii* population in Xiamen Bay, China. From 2006 to 2007, the "Minlongyu 2002" fishing boat was used, and four cruises were performed for seasonal sampling over two years. The vessel was a single-boat truss bottom trawler with the main engine power of 202 kW. The trawl net size was 41 × 26.8 m, and the mesh size of the cod-end was 20 mm. The surveys were conducted following the "Regulations for the Survey of Marine Fishery Resources (SC/T 9403-2012)". Towing took place approximately parallel to the coastline at a speed of 2.5 knots for 1 h. In 2016–2017, the "Minlongyu 62678" fishing boat was used, and seasonal sampling was conducted on four cruises. The vessel was a single-boat truss bottom trawler with the main engine power of 330 kW. The height of the bottom trawl net mouth was 2.5 m, the length of the trawl net was 24 m, the mesh of the cod-end was 20 mm, and the width of the truss was 27 m. Towing took place approximately parallel to the coastline at a speed of 2.5 knots for 1 h.

After the net was lifted, the entire catch was poured on the deck. First, the target species *J. belengerii* was picked from the catch. Thereafter, the fish were placed in marked plastic bags, and the samples were frozen for further analysis in a laboratory. The samples were thawed in the laboratory, rinsed, and counted, and the biological parameters were determined. Males and females were identified, and the total length (mm) and weight (g) were measured. A total of 1365 fish (663 in 2006–2007 and 702 in 2016–2017) were collected for this study.

### 2.3. Biological Characteristics

2.3.1. Growth

Data analysis was performed using Microsoft Excel and FAO-ICLARM Fish Stock Assessment Tools (FISAT II) [40,41]. Microsoft Excel was used to analyze the biology of the fish, whereas the FISAT II program was used to analyze the growth parameters ($L_\infty$, $K$, and $t_0$).

The length and weight of fish were analyzed using the Pauly equation [42] as follows:

$$W = a\,L^b \tag{1}$$

where $W$ is the weight (g), $L$ is the body length (mm), $a$ is the proportionality constant, and $b$ is the isometric exponent or slope indicating isometric growth, also known as the power exponent coefficient. $a$ can be regarded as a conditional factor for growth; the larger the $a$, the better the environmental conditions, such as the food base and hydrology. The value of $b$ is equal to 3 for the constant-growth-type fish. If the $b$ value is not equal to 3, it can be regarded as allometric growth.

Growth in length and weight was analyzed using the von Bertalanffy growth function (VBGF). The asymptotic length ($L_\infty$) and growth coefficient ($K$) were estimated using the input data from the length frequencies and the FISAT II program.

$$L_t = L_\infty\,[1 - e^{-K(t - t0)}] \tag{2}$$

$$W_t = W_\infty\,[1 - e^{-K(t - t0)}]^b \tag{3}$$

where $L_t$ and $W_t$ are the body length and body weight of t-year-old fish, respectively; $L\infty$ and $W\infty$ are the asymptotic length and asymptotic weight, respectively; K is the growth coefficient; t0 is the theoretical age when the fish body length is equal to 0.

The first and second derivatives of the VBGF were calculated to obtain the body length velocity equation (dL/dt) and acceleration equation ($d^2L/dt^2$). Similarly, the first

and second derivatives of the VBGF were calculated to obtain the body weight velocity equation (dW/dt) and acceleration equation ($d^2W/dt^2$), respectively.

### 2.3.2. Mortality

The instantaneous rate of total mortality ($Z$) was calculated by using the length-converted catch curve method (FISAT II software) [40,41]. The instantaneous rate of natural mortality ($M$) was obtained using Pauly's empirical formula [43]. The annual average surface water temperature T of the two periods in Xiamen Bay was $T_{2006}$ = 22.5°C and $T_{2016}$ = 22.3 °C, respectively.

$$\text{In } M = -0.0066 - 0.279 \times \text{In } L_\infty + 0.6543 \times \text{In } K + 0.463 \times \ln T \tag{4}$$

the instantaneous rates of fishing mortality ($F$) were calculated by subtracting the estimates of $M$ from $Z$ as follows:

$$F = Z - M \tag{5}$$

The exploitation rate ($E$) was calculated as follows:

$$E = F/Z \tag{6}$$

Then, the total mortality ($A$) was obtained using the following equation:

$$A = 1 - S \tag{7}$$

### 2.3.3. Feeding Intensity, Gender, and Gonad Maturity

The feeding intensity indicates the quantity of food in the stomach or intestine of the fish. Five grades were defined, namely, grade 0, empty stomach; grade 1, the food in the stomach is less than 1/2 of the stomach cavity; grade 2, the food in the stomach occupies 1/2 of the stomach cavity; grade 3, the stomach is full of food, but the stomach wall does not expand; Grade 4, the stomach is filled with food, and the stomach wall swells and becomes thinner.

Gonads were used to analyze maturity stages. Gonad maturity was judged according to the grade 6 visual grading method [44]. Six stages of gonad maturation were established for females and males: immature (I), maturing (II), mature (III), spawning (IV), spent (V), and resting (VI). Samples from different times of the year provided gonads at different developmental stages of the reproductive cycle. The sampling considered fish length to cover different sizes of *J. belengerii* within the maturity stage.

## 3. Results

### 3.1. Weight–Length Relationship

From 2006 to 2007, four field surveys were conducted to explore the fisheries biology of *J. belengerii* populations in Xiamen Bay, China. A total of 2707 fish were sampled, with a total weight of 48,882.3 g. In 2006, the average catch rate by weight was 1.75 kg/h, and the average catch rate by individuals was 97 ind/h. The body length ranged from 14 to 198 mm, and the weight varied from 0.06 to 163.84 g. The same four field surveys were conducted from 2016 to 2017. A total of 1784 fish were sampled, with a total weight of 46,426.5 g. In 2016, the average catch rate by weight was 1.93 kg/h, and the average catch rate by individuals was 94 ind/h. Body length ranged from 35 to 201 mm, and weight ranged from 0.8 g to 166 g.

The weight–length equations obtained in the two periods are as follows (Figure 2):

$$2006\text{-}2007: W = 7.80 \times 10^{-6} \times L^{3.21} \ (R^2 = 0.988, p < 0.01, n = 663).$$

$$2016\text{-}2017: W = 1.21 \times 10^{-5} \times L^{3.10} \ (R^2 = 0.976, p < 0.01, n = 702).$$

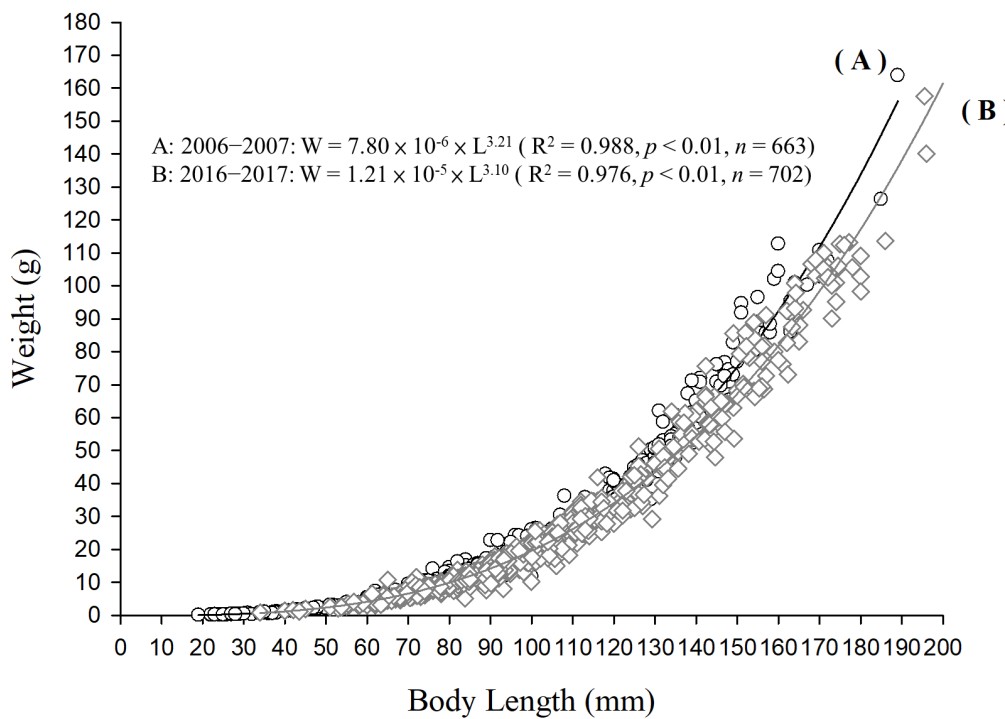

**Figure 2.** Relationship between body length and weight of *Johnius belengerii*: ((**A**): 2006–2007, (**B**): 2016–2017) in Xiamen Bay.

The growth condition factor $a$ was $7.8 \times 10^{-6}$ and the growth index $b$ value was 3.21 (2006–2007). The value of $b$ was not equal to three, which can be regarded as allometric growth. The growth condition factor $a$ was $1.21 \times 10^{-5}$ and the growth index $b$ value was 3.10 (2016–2017). The value of $b$ was also shown to be not equal to 3, which can be regarded as allometric growth.

There are two types of allometric growth patterns: positive ($b > 3$) and negative ($b < 3$). A positive allometric growth indicates that an increase in weight is dominant compared to increase in length, whereas a negative allometric growth indicates that an increase in length is more dominant than increase in weight. The $b$ values of the two periods 2006–2007 and 2016–2017 seem to imply that an increase in weight is more dominant than increase in length. In general, condition factors can be used to assess the physical and environmental conditions of a fish [45]. Upon comparing the condition factors between the two periods of 2006–2007 and 2016–2017, it was found that the $b$ value of the second period seemed to be greater than that of the first period. This means that the environmental conditions in the second period appeared to be better than those in the first period.

### 3.2. Growth Equation

In the period of 2006–2007, the growth coefficient ($K$) value was 0.70 year$^{-1}$, the asymptotic total length ($L_\infty$) could reach 199.5 mm, and the asymptotic body weight ($W_\infty$) could reach 178.5 g. According to Pauly's empirical formula, $t_0 = -0.50$. The VBGF is obtained as follows:

$$L_t = 199.5 \left[1 - e^{-0.70(t + 0.50)}\right]$$

$$W_t = 188.3 \left[1 - e^{-0.70(t + 0.50)}\right]^{3.21}$$

The first and second derivatives of the VBGF were calculated to obtain the body length velocity equation (dL/dt) and acceleration equation (d$^2$L/dt$^2$) as follows:
length velocity equation:

$$dL/dt = 139.65\, e^{-0.70(t + 0.50)}$$

length acceleration equation:

$$d^2L/dt^2 = -97.76 \ e^{-0.70(t + 0.50)}$$

Similarly, the first and second derivatives of the VBGF were calculated to obtain the body weight velocity equation (dW/dt) and acceleration equation (d²W/dt²) as follows:

weight velocity equation:

$$dW/dt = 423.11 \ e^{-0.70(t + 0.50)}[1 - e^{-0.70(t + 0.50)}]^{2.21}$$

weight acceleration equation:

$$d^2W/dt^2 = 296{,}18 \ e^{-0.70(t + 0.50)}[1 - e^{-0.70(t + 0.50)}]^{1.21}(3.21e^{-0.70(t + 0.50)} - 1)$$

When $d^2W/dt^2 = 0$, the inflection point of the weight growth age was obtained as $t_p = 1.17$.

In the period 2016–2017, $K = 0.550$ year$^{-1}$, $L_\infty = 220.5$ mm, $W_\infty = 222.5$ g, and $t_0 = -0.62$. The VBGF was obtained as follows:

$$L_t = 220.5[1 - e^{-0.55(t + 0.62)}]$$

$$W_t = 222.5[1 - e^{-0.55(t + 0.62)}]^{3.10}$$

The body length velocity equation (dL/dt) and the acceleration equation (d²L/dt²) are as follows:

$$dL/dt = 121.26 \ e^{-0.55(t + 0.62)}$$

$$d^2L/dt^2 = -66.70 \ e^{-0.55(t + 0.62)}$$

Similarly, the body weight velocity equation (dW/dt) and the acceleration equation (d²W/dt²) are as follows:

$$dW/dt = 379.36e^{-0.55(t + 0.62)}[1 - e^{-0.55(t + 0.62)}]^{2.10}$$

$$d^2W/dt^2 = 208.65e^{-0.55(t + 0.62)}[1 - e^{-0.55(t + 0.62)}]^{1.10}(3.10e^{-0.55(t + 0.62)} - 1)$$

When $d^2W/dt^2 = 0$, the inflection point of the weight growth age was obtained as $t_p = 1.44$.

By comparing the growth characteristics before and after a period of ten years, the inflection point age of the weight growth of *J. belengerii* showed a trend of increasing from 1.02 in the first period to 1.11 in the second period. In addition, the asymptotic body length $L_\infty = 220.5$ and weight $W_\infty = 222.5$ values of the second period were also larger than the first period values, $L_\infty = 199.5$ and $W_\infty = 188.3$. The growth coefficient $K = 0.550$ year$^{-1}$ in the second period was shown to be slower than $K = 0.70$ year$^{-1}$ in the first period.

These changes reflect the changes between the two periods. This also suggests that the driving force is the effectiveness of multiple long-term fisheries policies. Furthermore, this explains why the characteristics of fish growth gradually improved.

### 3.3. Mortality and Exploitation Rate

In this study, the period of 2006–2007 resulted in these values: $Z = 2.01$ year$^{-1}$, $M = 0.76$ year$^{-1}$, $F = 1.25$ year$^{-1}$, and the exploitation rate ($E$) = 0.62 year$^{-1}$. The period from 2016–2017 resulted in these values: $Z = 1.67$ year$^{-1}$, $M = 0.63$ year$^{-1}$, $F = 1.04$ year$^{-1}$, and $E = 0.62$ year$^{-1}$ (Table 1). According to Sparre and Venema (1992), the optimum exploitation ratio *Eopt* is 0.5, implying that the stock of *J. belengerii* in Xiamen waters seems to be overexploited.

**Table 1.** Biological characteristics and biological information values from two periods.

| Period | Biological Characteristics | Biological Information Value |
|---|---|---|
| 2006–2007 | Length range | 14–198 mm |
| | Weight range | 0.06–163.84 g |
| | Average body length | 91.77 mm |
| | Average body weight | 22.68 g |
| | Relationship of length and weight | $a = 7.8 \times 10^{-6}$; $b = 3.21$ |
| | Growth equation | $L\infty = 199.5$ mm; $W\infty = 188.3$ g; $K = 0.70$ |
| | Mortality coefficient | $Z = 2.01$; $M = 0.76$; $F = 1.25$; $E = 0.62$ |
| 2016–2017 | Length range | 35–201 mm |
| | Weight range | 0.8–166 g |
| | Average body length | 100.86 mm |
| | Average body weight | 25.58 g |
| | Relationship of length and weight | $a = 1.21 \times 10^{-5}$; $b = 3.10$ |
| | Growth equation | $L\infty = 220.5$ mm; $W\infty = 222.5$ g; $K = 0.55$ |
| | Mortality coefficient | $Z = 1.67$; $M = 0.63$; $F = 1.04$; $E = 0.62$ |

*3.4. Feeding Intensity*

In the spring of 2016, the male- and female-dominant body length groups accounted for 55.8% and 62.1% of the total, respectively. The dominant male and female weight groups accounted for 46.2% and 55.2% of the patients, respectively. In the 2006–2007 period, a total of 398 samples were collected and analyzed for gastrointestinal feeding intensity, including 60, 114, 17, and 207 individuals in spring, summer, autumn, and winter, respectively. Through the analysis of the degree of stomach contents, it was concluded that the feeding intensity in spring, summer, and winter was mainly level 1, and the feeding intensity in autumn was mainly level 2, accounting for nearly 50% (as shown in Figure 3).

In the 2016–2017 period, 671 samples were collected and analyzed for feeding intensity, including 101, 163, 178, and 229 individuals in spring, summer, autumn, and winter, respectively. It was concluded that the feeding intensity in summer was mainly level 1, with the largest proportion accounting for 47.9%; in spring and autumn, it was mainly level 2; and in winter, it was mainly level 0. The proportions of the three were between 30% and 40% (Figure 3).

*Johnius belengerii* feeds throughout the year, and seasonal variation in feeding intensity is evident [46]. In this study, the feeding intensity also varied seasonally, with a peak in summer and a low in winter.

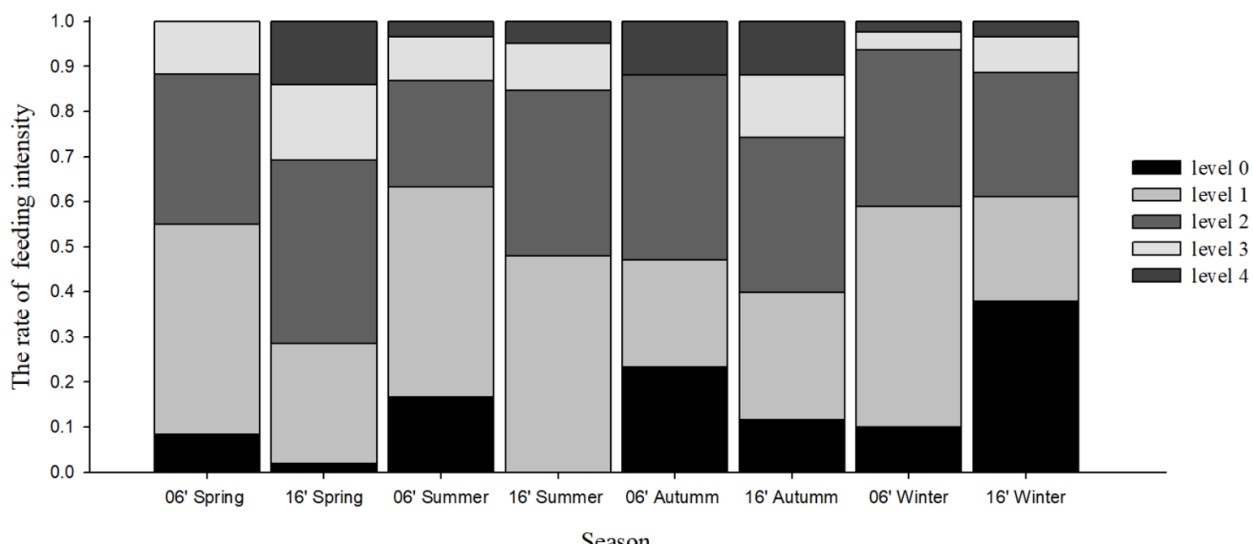

**Figure 3.** Seasonal proportions of feeding intensity of *Johnius belengerii* during the periods of 2006–2007 and 2016–2017.

### 3.5. Sex Ratio and Maturity

The sex ratio of females to males and the unknown ratio in different seasons for the two periods are shown in Table 2. In the 2006–2007 period, a total of 663 samples were collected and identified. The female-to-male ratio was 1.12:1. The ratio was higher in the summer and lower in the winter. In the 2016–2017 period, 702 samples were collected, and the ratio was 1.22:1. The ratio was also higher in the spring and lower in the winter. In the 2006–2007 period, the gonad maturity of both males and females was mainly stage II, accounting for 66.36% and 79.59% of the total number, respectively. In the 2016–2017 period, it was mainly stage II, accounting for 58.50% and 84.68% (Table 3; Figure 4).

**Table 2.** Sex ratio and unknown sex ratio in two periods.

| Period | Season | Total Number of Samples | Total Number of Females | Total Number of Males | Total No. of Unknown Sex | Sex Ratio (Female/Male) | Unknown Sex Ratio (Unknown Sex/Total Sample) |
|---|---|---|---|---|---|---|---|
| 2006–2007 | Spring (May) | 64 | 23 | 18 | 23 | 1.28 | 0.36 |
| | Summer (Aug.) | 186 | 21 | 16 | 149 | 1.31 | 0.80 |
| | Autumn (Nov.) | 189 | 18 | 17 | 154 | 1.06 | 0.81 |
| | Winter (Feb.) | 224 | 48 | 47 | 129 | 1.02 | 0.58 |
| | Total | 663 | 110 | 98 | 455 | 1.12 | 0.69 |
| 2016–2017 | Spring (May) | 103 | 52 | 29 | 22 | 1.32 | 0.21 |
| | Summer (Aug.) | 164 | 38 | 31 | 95 | 1.22 | 0.58 |
| | Autumn (Nov.) | 178 | 41 | 37 | 100 | 1.11 | 0.56 |
| | Winter (Feb.) | 257 | 72 | 69 | 116 | 1.04 | 0.45 |
| | Total | 702 | 203 | 166 | 333 | 1.22 | 0.47 |

In summary, in both 2006 and 2016, the proportion of males and females that could not be distinguished accounted for the vast majority, which means that most of the fish were immature individuals. Out of the individuals that could be distinguished in the four seasons of 2006, the fish samples were mostly females. In 2016, most of the samples, except in spring, were females, and the remaining three seasons were mostly males.

**Table 3.** Different stages of gonad maturation with the average body length and average body weight of *Johnius belengerii* in 2006–2007 and 2016–2017.

| Period | Gonadal Maturity Stage | I | II | | III | | IV | | V | |
|---|---|---|---|---|---|---|---|---|---|---|
| | | Unknown Sex | Female | Male | Female | Male | Female | Male | Female | Male |
| 2006–2007 | Number of samples | 455 | 73 | 78 | 29 | 12 | 5 | 8 | 3 | 0 |
| | Proportion of total number of samples (%) | 68.63% | 11.01% | 11.76% | 4.37% | 1.81% | 0.75% | 1.21% | 0.45% | 0 |
| 2016–2017 | Number of samples | 333 | 87 | 188 | 24 | 25 | 18 | 8 | 19 | 0 |
| | Proportion of total number of samples (%) | 47.44% | 12.39% | 26.78% | 3.42% | 3.56% | 2.56% | 1.14% | 2.71% | 0 |

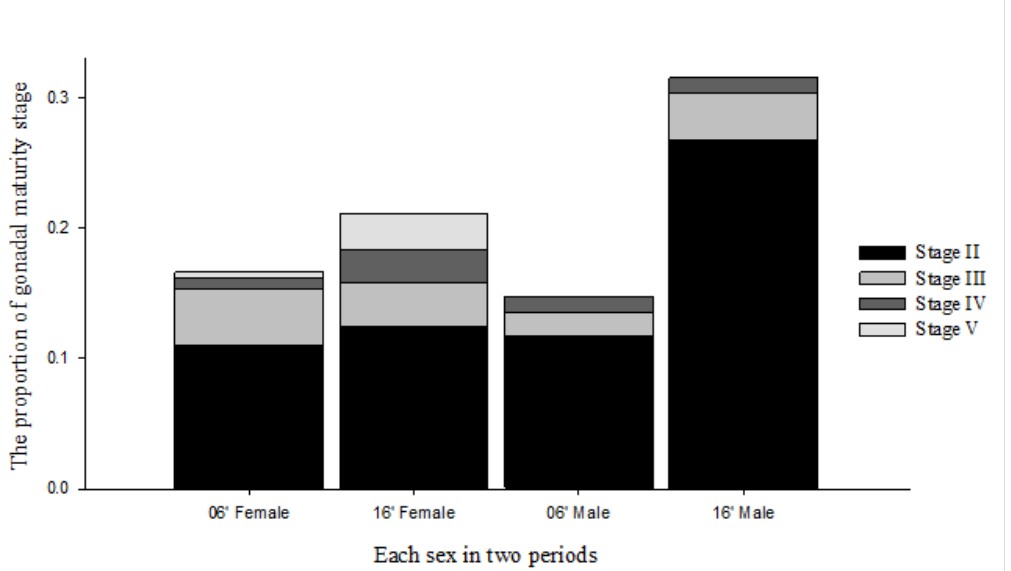

**Figure 4.** Proportion of gonadal maturity in females and males of *Johnius belengerii* in 2006 and 2016.

## 4. Discussion

### 4.1. Changes in Biological Characteristics

By comparing the two periods of 2006–2007 and 2016–2017, it is seen that the average body length increased from 199.5 mm to 220.5 mm, an increase of 9.09 mm and 9.91%. The average body weight increased by 2.90 g, an increase of 12.79%. This change showed that the fish's body length and body weight increased in the later period. The inflection point of body weight growth was 1.02 years old in 2006–2007, whereas in 2016–2017 it was 1.11 years old. The age at the inflection point increased, which indicates that the body weight of the fish tended to increase. Generally, the *b* value in the relationship between body length and weight can be used to determine whether a fish is growing at an isokinetic rate. Some studies have reported that most of the *b* values of marine fish, shrimp, and crabs are in the range of 2.5 to 3.5 [47]. In this study, the *b* values of *J. belengerii* were 3.21 and 3.10 during 2006–2007 and 2016–2017, respectively. The *b* values were not equal to three, indicating that both periods were allometric. Moreover, there were more immature individuals in 2006–2007 than in 2016–2017. Similarly, a is related to environmental factors such as fish feeding, habitat, and hydrological conditions. The larger a is, the more favorable the environmental conditions for fish growth [46]. In this study, a can also indicate the same situation, which was relatively larger at a later stage. We further compared the biological characteristics of *J. belengerii* in several water bodies in China (Table 4). Previous studies have only shown a relationship between body length and weight [26,29,48–51]. There have been relatively complete studies in the last ten years [30,52–54]. There was no consistent trend in the comparison of the different sea areas. However, the same waters in Xiamen exhibited a consistent trend. In Xiamen, they showed better biological characteristics during the later period.

From 1980 to 2000, fish miniaturization was widespread in China's marine fisheries [55]. Cases of small yellow croaker (*Larimichthys polyactis*) from three periods in the Lvsi fishing ground were shown. The average weight was 33.36 g in 1970, 83.96 g in 1980, and only 46.95 g in 1990 [56]. Liu et al. (2018) [57] also mentioned that the small yellow croaker has suffered from severe overfishing in the past 20 years, and its production collapsed and dropped sharply. Production in 2003 reached its lowest level, being only 18% of that in 1992. Subsequently, the resource gradually recovered, and by 2011 it exceeded the levels of the early 1990s [57]. The resistance and stability of the small yellow croaker to fishing and the environment are poor, thus the population structure is simple, and the individuals are miniaturized and young [56–58]. Li et al. [59] suggested that growth would

be better at relatively low fishing pressures and that allometric factors would also increase. In contrast, when fishing pressure was high, body length increased faster than body weight, indicating negative allometric growth [59].

**Table 4.** Growth and mortality parameters of *Johnius belengerii* in different seas of China.

| Location | Paper Publish Year | $a$ | $b$ | Asymptotic Length ($L\infty$) | Growth Coefficient (k) | t0 | Total Mortality (Z) | Natural Mortality (M) | Fishing Mortality (F) | Exploitation Rate (E) |
|---|---|---|---|---|---|---|---|---|---|---|
| Pearl River Estuary [48] | 1998–1999 | $3.44 \times 10^{-5}$ | 2.88 | | | | | | | |
| Beibu Gulf [49] | 2006–2007 | $1.67 \times 10^{-5}$ | 3.05 | | | | | | | |
| Min River Estuary [50] | 2006–2007 | $1.02 \times 10^{-5}$ | 3.16 | | | | | | | |
| Southwestern coast of Taiwan [51] | 2009–2010 | $4.44 \times 10^{-5}$ | 2.83 | | | | | | | |
| Ma'an Archipelago [26] | 2009–2010 | $1.35 \times 10^{-5}$ | 3.09 | | | | | | | |
| Liusha Gulf [29] | 2016–2017 | $9.02 \times 10^{-6}$ | 3.17 | | | | | | | |
| Fujian coastal area [30] | 2010–2011 | $8.47 \times 10^{-6}$ | 3.2 | 200 | 0.56 | −0.29 | 2.01 | 0.63 | 1.38 | 0.69 |
| Xiamen sea area [52] | 2015–2016 | $3.44 \times 10^{-5}$ | 3.08 | 220.5 | 0.56 | −0.61 | 2.25 | 0.63 | 1.62 | 0.72 |
| Xiamen Bay [53] | 2016–2017 | $1.14 \times 10^{-5}$ | 3.11 | 210 | 0.81 | −0.19 | 1.09 | 0.61 | 0.48 | 0.44 |
| Southern Zhejiang [54] | 2015–2018 | $5.63 \times 10^{-6}$ | 3.26 | 207.4 | 0.63 | −0.25 | 1.88 | 1.24 | 0.66 | 0.34 |
| Xiamen Bay [this study] | 2006–2007 | $7.8 \times 10^{-6}$ | 3.21 | 199.5 | 0.7 | −0.5 | 2.01 | 0.76 | 1.25 | 0.62 |
| Xiamen Bay [this study] | 2016–2017 | $1.21 \times 10^{-5}$ | 3.1 | 220.5 | 0.55 | −0.62 | 1.67 | 0.63 | 1.04 | 0.62 |

The mortality characteristics of this study are shown in Table 1. Meanwhile, the greater the mortality rate, the faster the decline in its resources [44]. Comparing the two periods shows that all coefficients became smaller in the late period. This result indicates that the fishing pressure is declining and *J. belengerii* is recovering slowly. Natural mortality also declined, indicating an improvement in the environment. We further compared the coefficients of several water sources in China (Table 4). Comparing the development intensity of the four sea areas, the Xiamen Bay, Fujian coastal area, and Xiamen sea area were similar, and the coastal area of southern Zhejiang was the lowest. The M value of the coastal waters of southern Zhejiang was higher than that of the other seas. Gulland believed that the optimal exploitation rate of fish resources is approximately 0.5 [60]. However, the values were greater than 0.5 in Xiamen waters. From this, it can be seen that the species has been in a state of overexploitation for the past ten years. To achieve sustainable resource utilization, it is necessary to strengthen relevant resource protection measures.

The phenomena of a decline in the natural sustainability of a species include a reduction in natural spawning activity, reduction in spawning population size, dramatic change in limited spawning grounds, and decrease in recruitment [61]. In this study, the sex ratios in 2006–2007 and 2016–2017 were 1.17:1 and 1.22:1, respectively, which were smaller than those in the Ma'an Archipelago (1.70:1) [26] and larger than those in the Xiamen waters (0.87:1) [52], Zhejiang southern coastal waters (0.91:1) [53], and Jiaozhou Bay (0.7:1) [62]. Generally, fish stocks increase the proportion of female individuals during periods of good living conditions (mainly nutritional conditions) to enhance the fecundity of the population. Conversely, if the number of males increases, the fecundity of the group decreases. The sex ratio of *J. belengerii* belongs to the former. However, in the case of deteriorating food conditions, some fish adopt a strategy of preferential maturation of females to ensure the continuation of the population [63]. In addition, on comparing the gonad maturation stages of the two periods, both stages I and II were dominant. In 2016–2017, the proportion decreased in stage I and increased in stages II and III. This indicated more spawning activity and larger spawning population sizes in later-spawning populations.

Previous studies have shown a correlation between feeding and water temperature changes in *J. belengerii*. In summer and autumn, the water temperature was higher and

the feeding intensity was relatively strong. However, in winter and spring, the water temperature was lower and the feeding intensity became weaker. In 2006–2007, the feeding intensity was mainly grade 1 in spring, summer, and winter, and grade 2 in autumn. However, in 2016–2017, the feeding intensity was mainly grade 2 in spring and autumn, grade 1 in summer, and grade 0 in winter. It can be seen from the above that *J. belengerii* ingested food in spring, summer, and autumn, whereas food intake in winter was less or even non-existent.

*4.2. Effectiveness of Fisheries Policies*

Realizing the sustainable development of marine fishery resources is a difficult problem worldwide. Over the past two decades, most countries have recognized the need for improved fisheries management. Similar to other countries, marine fish in China are mostly unprotected and unassessed stocks, or at least there are no public assessment data. Generally, fisheries policies determine management priorities based on fisheries assessments, such as maximum sustainable yield, biodiversity, maximizing profitability, and fishing production or levels [64]. In China, the government is increasingly aware of the need to further improve the regulatory framework to achieve greater economic, social, and environmental sustainability and resilience [14,15,65]; in particular, the impacts of poor management, the overexploitation of marine ecosystems, and climate change need to be addressed [66]. Delaval et al. [67] described ongoing and future research and conservation efforts using case studies of four species with varying conservation statuses. When properly managed and integrated with other sectors, fisheries can feed an expanding population [68]. It is difficult to identify the effectiveness of a particular policy. The catch data in the region are generally of low resolution, and there is a dearth of fishing efforts and socio-economic data, making quantitative and qualitative assessments of the impacts on the fisheries sector very difficult. Furthermore, it remains challenging to separate the impacts from other measures and factors, particularly given their synergistic effects [66]. Fisheries policies and laws are typically developed at the national level. Fisheries management is often decentralized at the local or provincial levels. However, fisheries policy does not occur in a vacuum; it interacts at multiple levels with a range of other legal and policy instruments [68]. Over the past decade, the fisheries sector has particularly increased its focus on resource and ecosystem sustainability, which has led to the wider use of stock management policies [64]. Policies and practices are designed to prevent illegal, unreported, and unregulated policies. Therefore, long-term fishery resource assessment is crucial, such as the long-term changes in biological information in this study. Effective policy design can be examined through case studies. It can provide an important basis and visibility for effective management.

The Marine Environmental Protection Law was promulgated in 1982. Similarly, the Fisheries Law has been promulgated as a fisheries management policy since 1986. This law provides a legal basis for the development and protection of fishery resources and fishermen's legitimate rights and interests. From 1986 to 1996, the average annual growth rate of marine fishing production reached 11.8% and peaked at 13.3 million tons in 1998. More than 250,000 marine fishing vessels were registered in 1992. Unfortunately, fishery resources have already begun to deplete. From 1990 to 2010, the marine economy became the fastest growing area of China's economy, contributing approximately 10% of the annual GDP. The gross production value of the marine industry accounts for 60% of the national GDP and 90% of the total imports and exports. Undoubtedly, the vigorous development of coastal areas and rapid economic growth of the ocean have put enormous pressure on marine ecosystems.

Faced with overfishing and continuous degradation of marine ecosystems, the government has adopted a series of management measures for marine fisheries. Five-year plans from the 21st century have further encouraged investment in marine aquaculture and distant-water fishing vessels. Fisheries policies were successively introduced and implemented in the "Twelfth Five-Year Plan" period (2011–2015). These measures include input and output controls, technical limitations on gear size and operation types, closed

areas and periods, ecological measures, and economic incentives. One of the most dramatic changes in fisheries management was the introduction of a fishing moratorium in the mid-1990s. Su et al. (2021) [65] mentioned that before 2011, China's fisheries policy mainly focused on the input control of fishing gear, fishing vessels, and fishermen. Since 2011, China's fisheries policy has been adjusted to control the production in the past decade. The management of fishery resources is gradually changing from input control to input-output control. In addition to the fishing moratorium, the Ministry of Agriculture proposed a "zero growth" policy in 1999 and a "negative growth" policy in 2000 [66]. The policy change meant that the state no longer encouraged increased fishing production, in stark contrast to the past, with production growth as a measure of government performance.

The "China Ocean Agenda 21" was announced to clarify the basic strategies, strategic goals, countermeasures, and main action areas for the sustainable development of the ocean [69]. A series of five-year plans have placed more emphasis on resource conservation, environmental protection, and ocean awareness [1]. To restore fishery resources, all coastal provinces have implemented restocking, stock enhancement, and stock replenishment of fishery resources [5]. A landmark example is the large-scale shrimp breeding and release of *Fenneropenaeus chinensis* in the Bohai Sea that began in 1984 [70,71]. By 2008, more than 100 species had been stock enhanced and released, including fish, crustaceans, and shellfish, and about 20 billion seedlings were put in each year [70,72]. Stock enhancement and release of lancelets (*Branchiostoma japonicum* and *B. belcheri*) were also performed in Xiamen, Fujian [73]. These aquatic organisms are the links between invertebrates and vertebrates. Saying it is a "fish" does not make it a fish. Its shape is similar to that of a small fish, with a body that is flat on the side, a length of about 3–5 cm, and it is translucent with a pointed head and tail; additionally, it lives in the coastal sediment [74–76]. Lancelets have been listed as second-class protected animals in China [74–76]. In addition to lancelets, since 2003, the Xiamen municipal government has carried out stock enhancement and release for a total of 16 species.

Different measures such as "marine protected areas," "marine reserve," "no-take zone", or "fish box" are mostly considered important tools for managing and protecting fishery resources [77]. Traditionally, MPAs and reserves, including specific fisheries management measures such as closures and fishing restrictions, have benefited fisheries through stock enhancement and management [78]. The core "no-take" role of MPAs can play an important role in reversing the decline in local fish population and productivity. Habitat protection is important for life cycle stages, including spawning, larval colonies, nursery grounds, and primary feeding grounds. The Xiamen Rare Marine Species National Nature Reserve was established in 2000, and the protected species included lancelets and *Sousa chinensis*. Development and utilization activities in the protected area are subject to a permit system, including scientific research, teaching, inspection, and various development and utilization activities engaged in production and operation. Fortunately, this measure indirectly protected other fishery resources.

In 2015, a new marine spatial planning (MSP) scheme was proposed, called the "Marine Ecological Red Line" (MERL) [38,79–81]. It aimed to protect ecologically sensitive areas and important ecological functions, and formulate reasonable development boundaries and industrial layouts. Although "Redline" policies have been proposed for several years, the technical and practical approaches to implementing them are still limited [82]. Inadequate ecological information and inappropriate zonings are major constraints and challenges in MSP and management processes [38]. The redline area refers to the degree to which ecosystem disturbances reflect changes in human activity and the natural environment. This represents high ecological sensitivity and indicates that these areas have experienced species loss and ecological diversity deterioration in the past [38]. Therefore, the sustainable use of coastal resources can be promoted through relevant conservation and management strategies [83–85].

Changes in biological information are closely related to a series of fisheries management and environmental strategies, such as setting up closed fishing periods and

non-fishing areas and establishing fisheries restoration in marine protected areas. The fisheries policies pursued during this period can be seen to have contributed significantly to the recovery of resources. The results of our study support the implementation of this period. In the future, fisheries management will face many policy challenges, such as multi-species fisheries, long coastlines, numerous fishing ports, a large proportion of small fishing vessels, and scattered landing sites for catches [65]. On the premise of maintaining the livelihood of fishermen and the supply of aquatic products, it cannot be overemphasized to attach importance to the restoration of coastal fishery resources and improvement of the marine ecological environment. However, a lack of biological knowledge often leads to misconduct in fisheries protection. Establishing appropriate fisheries policies, such as conservation actions and marine protected area networks, relies on a fundamental understanding of the demographics, population distribution, migration behavior, and habitat requirements of different species [67]. Delpeuch and Hutniczak [64] proposed recommendations to facilitate future policy change through better use of data, commitment mechanisms, non-sectoral policies, and consultation processes. Overall, China's marine fisheries policy has focused on the sustainable development of fishermen's livelihoods, habitat restoration, and offshore fishery resources in the last decade. Our research appears to be effective and beneficial for assessing long-term changes in biological information.

## 5. Conclusions

China's rapid economic development has put enormous pressure on marine ecosystems, leading to the overfishing or extinction of marine fish populations. Since 1986, the Fisheries Law has been enacted as a fisheries management policy. In the past two decades, a series of five-year plans have been successively introduced and implemented. Fisheries measures, including input and output controls, technical limitations on gear size and type of operation, closed areas and periods, ecological measures, and economic incentives have been implemented to improve resource decline. Long-term fishery resource assessments are critical, as changes in biological information can provide important evidence and visibility for effective management. Changes in biological characteristics indicate an improvement in the later period, which is closely related to a series of fisheries management strategies, such as setting closed fishing periods and non-fishing areas and establishing fishery restoration marine protected areas. The results of our study reveal the effectiveness of a long-term series of fisheries policies.

**Author Contributions:** Conceptualization, L.-M.H., J.-Q.W. and T.-J.C.; methodology, L.-M.H., J.-Q.W., T.-J.C. and Y.-J.S.; software, J.-Q.W., Y.-J.S. and T.-J.C.; validation, L.-M.H., J.-Q.W., T.-J.C. and Y.-J.S.; formal analysis, J.-Q.W., T.-J.C. and Y.-J.S.; investigation, L.-M.H., J.-Q.W., Y.-J.S., T.-J.C. and J.L.; resources, L.-M.H., J.-Q.W., J.L. and Y.-J.S.; data curation, T.-J.C. and Y.-J.S.; writing—original draft preparation, L.-M.H., T.-J.C. and Y.-J.S.; writing—review and editing, T.-J.C. and Y.-J.S.; visualization, L.-M.H., J.-Q.W., T.-J.C., Y.-J.S. and L.-M.H.; supervision, L.-M.H., J.-Q.W. and T.-J.C.; project administration, L.-M.H.; funding acquisition, L.-M.H., T.-J.C. and Y.-J.S. All authors have read and agreed to the published version of the manuscript.

**Funding:** This work was supported by the Jimei University grant No. ZQ2019041, Jimei University grant No. C619061, Jimei University Doctoral Research Start-up Fund (C612012) and the Natural Science Foundation of Fujian Province, China (2021J01825). The funders had no role in study design, data collection and analysis, decision to publish, or preparation of the manuscript.

**Institutional Review Board Statement:** Not applicable.

**Informed Consent Statement:** Not applicable.

**Data Availability Statement:** Not applicable.

**Acknowledgments:** All authors would like to thank Editage for providing language help, and to the reviewers for their constructive criticism and improvement of the manuscript.

**Conflicts of Interest:** The authors declare no conflict of interest.

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
