# Peer review of "Revealing the Effectiveness of Fisheries Policy: A Biological Observation of Species Johnius belengerii in Xiamen Bay"

_jmse, doi:10.3390/jmse10060732_

Round 1

Reviewer 1 Report

  • The introduction is too long and has to be shortened
  • English must be improved
  • The growth study was based on the analysis performed using Microsoft Excel and FISATII, why didn't you use the otoliths reading?

Author Response

Reviewer #1: Comments to jmse-1731352 by Huang et al. (please, also see annotated manuscript):
We are much grateful for your careful reading of our manuscript and your valuable comments and suggestions to help improve the paper. We have now carefully revised the paper in light of all the comments and suggestions. The following is a point-by-point response.

  1. The introduction is too long and has to be shortened.

Answer: We have followed your comment and have removed and corrected redundant statements.

  1. English must be improved.

Answer: We have followed your comments. We have sought the help of professional English editors before submitting the manuscript. Attached is the English editorial note. If the reviewers still feel the need for reinforcement, we will again seek help from the English editor recommended by the journal.

  1. The growth study was based on the analysis performed using Microsoft Excel and FISATII, why didn't you use the otoliths reading?

We have followed your comments. We understand that the otolith is the most commonly used structure for determining the age of fish. To determine the fish age (using scales and otoliths), there are several protocols that help you to see better annual growth rings. This tool has been widely used in age judgment. Then, on this basis it can make age group determinations and mortality estimates. FiSAT II is a program package developed mainly for the analysis of length-frequency data, but also enables related analyses, of size-at-age, catch-at-age, selection and other analyses. Likewise, we consider FiSAT II. It is also one of the widely used population assessment tools recommended by FAO-ICLARM.

Reviewer 2 Report

v

This paper is well-written and presents a coherent argument – that more restrictive government policies have improved the stock of Belangers croaker in Xiamen Bay in China. However, I have the following three queries for the authors.  

(1) What is the evidence to support your main assertion that the improved stock of Belangers croaker was caused by changed government fisheries management policies?

You write that:

two sets of biological information from 2006-2007 and 2016-2017 were compared to evaluate the long-term effectiveness of a series of fishery policies” (lines 113-115)

The changes in biological characteristics show that there is a phenomenon of improvement in the later period, which are closely related to a series of fishery management strategies, such as setting closed fishing periods and non-fishing areas, and establishing fishery restoration marine protected areas. This result seems to reveal the effectiveness of a long-term series of fisheries policies”. (lines 18-22)

“These changes reflect the changes in the two periods. This also suggests that the driving force is the effectiveness of multiple long-term fishery policies. Furthermore, this explains why the characteristics of fish growth gradually improved” (lines 298-300)

The fishery policies pursued during this period can be seen to have contributed significantly to the recovery of resources” (lines 525-527)

Changes in biological characteristics indicate an improvement in the later period, which is closely related to a series of fishery management strategies, such as setting closed fishing periods and non-fishing areas and establishing fishery restoration marine protected areas. The results of our study reveal the effectiveness of a long-term series of fishery policies” (lines 552-556).

 But how can you be certain that the improvement in the fish stocks between 2006-7 and 2016-2017 can be attributed to the change in fisheries policies? You yourselves acknowledge how difficult it is to demonstrate the effectiveness of fisheries policies:

It is difficult to identify the effectiveness of a particular policy. The catch data in the region are generally of low resolution, and there is a dearth of fishing effort and socio-economic data, making quantitative and qualitative assessments of the impacts on the fishery sector very difficult. Furthermore, it remains challenging to separate the impacts from other measures and factors, particularly given their synergistic effects [67]. Fisheries policies and laws are typically developed at the national level. Fisheries management is often decentralized at the local or provincial levels” (lines 438-445).

So what guarantee can you give that there is a causal, rather than simply a correlative or coincidental, link between fisheries policies and improved stock levels of Belangers croaker in Xiamen Bay between 2006 and 2017?

(2) Your Results seem contradictory: on the one hand you say the stock of Belangers croaker has improved between 2006 and 2017:

The changes in biological characteristics show that there is a phenomenon of improvement in the later period” (lines 18-19)

Yet on the other hand you say the stock has declined:

Comparing the two periods, all coefficients became smaller in the late period. This result indicates that J. belengerii is declining under fishing pressure and is recovering slowly” (lines 388-390)

From this, it can be seen that the species has been in a state of overexploitation for the past ten years. To achieve sustainable resource utilization, it is necessary to strengthen relevant resource protection measures” (lines 397-399)

(3) What do the terms “changing industries” and “dual controls” (lines 67) mean?

Author Response

Reviewer #2: Comments to jmse-1731352 by Huang et al. (please, also see annotated manuscript):

We are much grateful for your careful reading of our manuscript and your valuable comments and suggestions to help improve the paper. We have now carefully revised the paper in light of all the comments and suggestions. The following is a point-by-point response.

  1. This paper is well-written and presents a coherent argument – that more restrictive government policies have improved the stock of Belangers croaker in Xiamen Bay in China.

Answer: We really appreciate some very good and encouraging comments from you.

  1. What is the evidence to support your main assertion that the improved stock of Belangers croaker was caused by changed government fisheries management policies?

You write that:

two sets of biological information from 2006-2007 and 2016-2017 were compared to evaluate the long-term effectiveness of a series of fishery policies” (lines 113-115)

The changes in biological characteristics show that there is a phenomenon of improvement in the later period, which are closely related to a series of fishery management strategies, such as setting closed fishing periods and non-fishing areas, and establishing fishery restoration marine protected areas. This result seems to reveal the effectiveness of a long-term series of fisheries policies”. (lines 18-22)

These changes reflect the changes in the two periods. This also suggests that the driving force is the effectiveness of multiple long-term fishery policies. Furthermore, this explains why the characteristics of fish growth gradually improved” (lines 298-300)

The fishery policies pursued during this period can be seen to have contributed significantly to the recovery of resources” (lines 525-527)

Changes in biological characteristics indicate an improvement in the later period, which is closely related to a series of fishery management strategies, such as setting closed fishing periods and non-fishing areas and establishing fishery restoration marine protected areas. The results of our study reveal the effectiveness of a long-term series of fishery policies” (lines 552-556).

But how can you be certain that the improvement in the fish stocks between 2006-7 and 2016-2017 can be attributed to the change in fisheries policies? You yourselves acknowledge how difficult it is to demonstrate the effectiveness of fisheries policies:

It is difficult to identify the effectiveness of a particular policy. The catch data in the region are generally of low resolution, and there is a dearth of fishing effort and socio-economic data, making quantitative and qualitative assessments of the impacts on the fishery sector very difficult. Furthermore, it remains challenging to separate the impacts from other measures and factors, particularly given their synergistic effects [67]. Fisheries policies and laws are typically developed at the national level. Fisheries management is often decentralized at the local or provincial levels” (lines 438-445).

So what guarantee can you give that there is a causal, rather than simply a correlative or coincidental, link between fisheries policies and improved stock levels of Belangers croaker in Xiamen Bay between 2006 and 2017?

Answer: We have followed your comments. As in my title "Revealing the effectiveness of fisheries policy: a biological observation of species Johnius belengerii in Xiamen Bay", we infer from a biological observation of species Johnius belengerii. Likewise, we describe in the Introduction, "The assessment of J. belengerii in Xiamen waters, which included investigation of parameters such as sex ratio, length-weight analysis, growth parameters, level of exploitation (natural mortality rate, fishing mortality rate, and total mortality rate), and feed intensity.”. Bioinformation changes between the two period were also seen in the results.

  1. Your Results seem contradictory: on the one hand you say the stock of Belangers croaker has improved between 2006 and 2017:

The changes in biological characteristics show that there is a phenomenon of improvement in the later period” (lines 18-19)

Yet on the other hand you say the stock has declined:

Comparing the two periods, all coefficients became smaller in the late period. This result indicates that J. belengerii is declining under fishing pressure and is recovering slowly” (lines 388-390)

From this, it can be seen that the species has been in a state of overexploitation for the past ten years. To achieve sustainable resource utilization, it is necessary to strengthen relevant resource protection measures” (lines 397-399)

Answer: We have followed your comments and corrected this error.

We fixed the "Comparing the two periods, all coefficients become smaller in the later period. This result indicates that the fishing pressure is decreasing and J. belengerii is slowly recovering" (lines 389-390)

  1. What do the terms “changing industries” and “dual controls” (lines 67) mean?

Answer: We have followed your comments and corrected this error.

We fixed the “fishermen changing jobs, controls of fishery inputs and outputs,” Line 68-69.